# A Comparison of Neighborhood-Scale Interventions to Alleviate Urban Heat in Doha, Qatar

**Salim Ferwati** [1] **, Cynthia Skelhorn** [1,*], **Vivek Shandas** [2] **and Yasuyo Makido** [2]

1   Department of Architecture and Urban Planning, Qatar University, Al Dafna, Doha 2713, Qatar;
    sferwati@qu.edu.qa
2   Toulan School of Urban Studies and Planning, Portland State University, 1825 SW Broadway, Portland,
    OR 97201, USA; vshandas@pdx.edu (V.S.); ymakido@pdx.edu (Y.M.)
*   Correspondence: cynthskelhorn@gmail.com

**Abstract:** Recent evidence suggests that many densely populated areas of the world will be uninhabitable in the coming century due to the depletion of resources, climate change, and increasing urbanization. This poses serious questions regarding the actions that require immediate attention, and opportunities to stave off massive losses of infrastructure, populations, and financial investments. The present study utilizes microclimate modeling to examine the role of landscape features as they affect ambient temperatures in one of the fastest growing regions of the world: Doha, Qatar. By modeling three study sites around Doha—one highly urbanized, one newly urbanizing, and one coastal low-density urbanized—the research indicates that at the neighborhood scale, the most effective scenario was that of adding mature trees along the sides of roads. In the coastal study area, the model results estimated a maximum hourly air temperature reduction of 1.35 °C, and in the highly urbanized inland site, surface temperature reductions were up to 15 °C at 12:00. While other scenarios were effective at reducing air and surface temperatures, the mean radiant temperature was also increased or nearly neutral for most of the other scenarios. This result highlights the need to develop improved shading measures for pedestrian pathways and outdoor recreational areas, especially for highly urbanized inland areas in Doha and cities with similar climatic conditions.

**Keywords:** urban climate; urban microclimate; microclimate modeling; urbanization; sustainable development; neighborhood adaptation; climate change

## 1. Introduction

Warnings of climate change impacts are proclaimed on a far more regular basis [1], while at the same time, the percentage of populations residing in urban areas is steadily increasing. In 2014, 54% of the world's population lived in urban areas, and this is expected to rise to 66% by 2050, with the most rapid urbanization in Asia and Africa [2]. Urbanization often leads to greater urban density, both in terms of population and the physical built environment. The implications of climate change on cities is still emerging, though a recognition that the interaction of social, infrastructure, and ecosystems features can amplify disasters that are borne from a warming planet is consistent [3].

In fact, recent evidence suggests that many densely populated areas of the world will be uninhabitable in the coming century due to the depletion of resources, climate change, and increasing urbanization leading to changes in the microclimate that directly affect livability [4,5]. One immediate and notable impact on cities is the phenomena of the urban heat island (UHI) effect, whereby measurable differences in both air and surface temperatures are often found between an urban area and surrounding rural areas [6,7]. A range of factors, including the city size, as measured by population, the increased density of human-made structures, and surface materials that are

drier than their surroundings and radiate sensible heat, and anthropogenic sources of heat, such as waste heat from vehicles and buildings, are known to contribute to the UHI [8–10]. An increasing number of cities already experience temperature and humidity ranges that challenge the ability of residents to find acceptable levels of comfort outdoors during summer, and even the hotter periods of the day in milder seasons, such as spring and autumn. A recent study of future temperatures in Southwest Asia found that many cities in the region of the Arabian Gulf will become uninhabitable by 2100 under current climate change projections [5], while a Lancet article detailed the population–environment–development dynamics that require an urgent focus on survivability for the Arab world [11]. The potential for some populous regions of the world to become uninhabitable poses serious questions about actions that require immediate attention, and opportunities to stave off massive human population health impacts.

A growing body of literature emphasizes the role of the alteration of the built environment as a strategy for heat mitigation [12–16]. Commonly proposed interventions include tree planting, the use of green roofs, and an overall increase in green spaces [17–19], as well as lightening roads, roofs, and buildings to increase albedo [20,21], all of which are increasingly popular in combating extreme heat. While such studies are helpful in guiding initial heat mitigation efforts, they are quite general, and face several challenges that make them not wholly useful in practice. For example, areas vary in the composition and configuration of landscape features, and recommending greening in an industrial area that contains little planting spaces may prove challenging due to the lack of available planting pace. Similarly, studies of heat mitigation in the built environment often refer to urban areas as homogeneous units, with limited differentiation between intra-urban land cover types [14], making the application of mitigation techniques difficult or ineffective at best. To guide the practice of urban heat mitigation efforts, practitioners need models that: (1) are adaptable to distinct cities and land uses, providing thresholds of mitigation potential; and (2) describe land cover types within the city, and thus indicate a variety of best intervention strategies, rather than one generalized approach.

The present study examines the role of specific landscape features as they affect ambient temperatures in one of the fastest growing regions of the world: Doha, Qatar. The city of Doha offers several advantages to examining the application of heat-reducing strategies, including the average minimum and maximum temperatures ranging from 13 °C to 42 °C (from cool to extreme hot), and can exceed 50 °C as an hourly maximum during the summer. Additionally, recent research about Doha's urban climate suggests that the land-use and land-cover change are likely increasing ambient temperatures [22,23], which further amplifies emerging livability challenges in the region [24,25]. We build on an emerging literature about thermal comfort and livability in the Middle East by asking three questions. (1) To what extent are conventional treatments for cooling neighborhoods effective in an arid desert climate? (2) What strategies show the most promise for cooling the ambient temperature? (3) How do similar cooling treatments vary in their effectiveness across diverse land-use and land-cover conditions in the city? We address these questions through the use of empirical data, which were collected over the course of three years, and the application of state-of-the-art fluid dynamic modeling systems. To contextualize the study, we begin by fleshing out a few of the main challenges facing the region, and then describe our methods for addressing these questions.

With the extremely hot temperatures in Qatar generally, and Doha specifically, planners, engineers, and architects must consider the potential impacts of climate change. The Lancet, Britain's premier health journal, calls climate change "the biggest global health threat of the 21st century" [26]. Qatar is predicted to have 65 days of heat wave conditions per year by 2020 under the RCP8.5 (Relative Concentration Pathway 8.5) (business-as-usual) modeling scenario, with this number expected to increase to 88 by 2050 [27]. Exposure to hot days may result in heat stroke (or, if directly in the sun, it is referred to as sun stroke), which is a life-threatening condition when a person collapses suddenly due to overheating [28]. A number of other direct and indirect impacts have been studied, including the increased risk of cardiovascular disease, worsening of respiratory conditions, risk of decreased kidney function, risk of adverse birth outcomes, and the changes in distribution of vector-borne disease [29].

Also, reduced labor productivity is already being seen in Gulf countries and elsewhere, which has a direct impact on both economic and well-being outcomes.

Some of the urban heat can be tempered through urban design, yet as Qatar has grown, little attention has been given to the implications of climate change. As an arid peninsula projecting into the Gulf, Qatar has approximately 563 km of coastline, and until recently was primarily made up of small fishing and pearling settlements. Since the late 1940s, Doha, the largest city and capital of Qatar, has experienced stages of urban growth ranging from traditional light-colored, low-rise, densely built neighborhoods to ambitiously sprawling low-density neighborhoods and a coastal center of high-rise towers through the implementation of small to mega-scaled transformative development projects. While much of Doha's built environment is designed similar to cities across the northern hemisphere, what has taken many centuries to evolve in other places has happened in less than half a century in Doha, and has been noted often as lacking the integration between the master planning and implementation phases that might have otherwise led to an overall coherent city plan [24]. With increasing concerns about climate change impacts and urbanization for such an extreme desert climate as that of Qatar, government planners and environmental managers in Doha and other Qatari cities are examining strategies to modify its further growth with a shift to more sustainable and livable results [30,31].

The existing built environment and projections of future changes to ambient temperatures in Doha pose several challenges for adapting to climate change. Climate models suggest that summer heat will be intensified by the UHI effect, with recent research noting increasing trends of warm temperature extremes [25]. Furthermore, a study specific to climate change projections in the Gulf region predicts that under a business-as-usual scenario of future greenhouse gas concentrations, many areas of the GCC region are likely to become uninhabitable due to intolerable rises in the wet-bulb temperature [5]. For Doha, the combined effects of a highly built out urban landscape, an expectation for increasing temperatures, and a rapid urbanization create an acute need for rapid improvement in urban design practices.

We propose an examination of Doha's temperature, humidity, and thermal comfort conditions in neighborhoods containing varying land-use and land-cover conditions. Considering the distinction between the urban canopy layer (UCL), which is ground height to approximately roof height, and urban boundary layer (UBL), which extends above roof level [32,33], we examine the microscale relationships between land cover, building materials, and spatial variations in three study areas. The aim of the present study is to understand which physical features of urban design can improve thermal comfort in outdoor spaces during periods of intense heat. We propose that the intensity of the near-surface air temperature is correlated with the density and the distribution of certain impervious surfaces and materials, and that the selective change of these surfaces and materials can improve thermal conditions and thermal comfort.

## 2. Materials and Methods

To address this research aims of the present study, we draw on several datasets and modeling systems. The empirical datasets are derived from weather stations that the authors distributed throughout the study region. A total of 10 weather stations were assembled and stationed in areas representing diverse physical geographies and built environments. The weather stations use solar power to track temperature, humidity, and windspeed. Tethered to an Omega datalogger, each weather station samples each parameter at 10-minute increments.

The weather station data were instrumental in calibrating the modeling system, ENVI-met (Essen, Germany). ENVI-met™ (V4.1.3, Winter Release 2016–2017) is a complex fluid dynamic model that uses a series of input parameters for each site to provide a spatially explicit description of the distribution of microclimates. The model has been developed and regularly improved since 1998 [34,35], and was selected because it is specifically designed for investigating changes to landscape and the built environment in urban areas.

The model uses a computational fluid dynamics (CFD) approach, employing the Reynolds-averaged non-hydrostatic Navier–Stokes equations for the wind field, the k-epsilon turbulence model, and a combined advection–diffusion equation with the alternating-direction implicit (ADI) solution technique [36] to model the interaction between microclimate and urban surfaces, such as walls, pavements, and vegetation. It employs both a soil model, with soil temperature calculated for natural soils and for artificial seal materials down to a depth of $-4$m, and a vegetation model, which allows complex three-dimensional (3D) vegetation geometries and accounts for heat and vapor exchanges with the atmosphere. ENVI-met uses the finite difference method to solve the numerous partial differential equations (PDE) [35]. Spatial resolution can be selected in the range of 0.5 to 10 m. While the model is three-dimensional, our model components are currently placed on a flat surface terrain, which is representative of the terrain in Qatar.

ENVI-met requires two main input files: a configuration file, which contains settings for initialization values and timings; and an area input file, which allows the user to design the layout of the site to be modeled, including the location and types of buildings, trees, and other vegetation, surfaces (e.g., asphalt and concrete), and soils. Areas to be modeled are digitized on a rectangular grid with the vertical top of the model set at a maximum of 2500 m, and a surface area of up to $250 \times 250$ cells. Data are input on a cell-by-cell basis. Points that are of interest for careful examination (e.g., for corresponding field measurements) can be specified as receptor points. The timing of model runs is set by the user, with a typical model simulation starting from 06:00, simulating a period of 12–24 h, and saving data once per hour (or more frequently, as necessary). Due to the extensive calculations and numerous outputs, the time required to complete one simulation period can be 24–48 h or more depending on the size of the model area, spatial resolution, and total number of hours simulated [37].

### 2.1. Site Selection

The overall study of climate adaptation within which this piece of research was performed had installed a total of nine weather stations in the vicinity of the capital city of Doha, with access to a 10th station at Qatar University. We selected three neighborhoods from these weather station locations (Figure 1) that were diverse in terms of their geographic distribution and land-use characteristics, but also representative of several styles of development currently found. The selected neighborhoods can be characterized as: urbanized—the area surrounding weather station five in Al Waab; coastal—surrounding weather station nine in Al Khulaifat; and developing—surrounding weather station one in Umm Salal Mohammad. Based on the availability of robust weather station data, the model was parameterized for a summer day in September 2014, using the urban microclimate model ENVI-met V4.

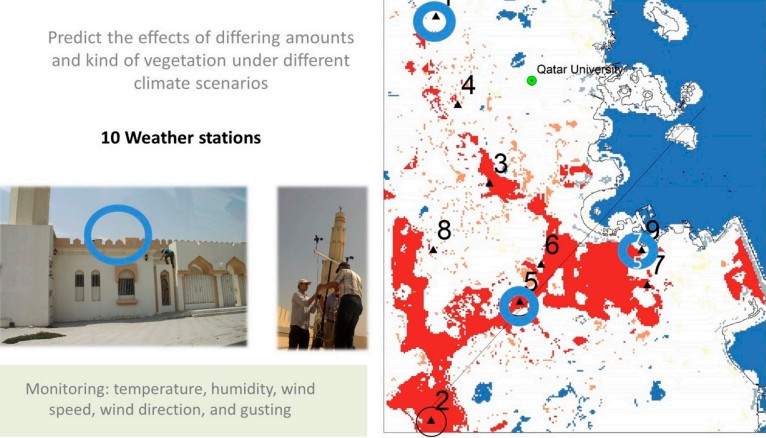

**Figure 1.** Locations of three case study areas, selected from neighborhoods in the vicinity of 10 weather stations.

Since each site will undergo specific heat mitigation treatments, we note several features that require further explication. The developing site (site one) in Umm Salal Mohammad was approximately one-third of the site that was developed at the time of the study, with a mosque in the center and one compound of two-story villas in the northern corner of the site, with a few more compound villas along the southern edge of the site. The remainder of the site was bare soil.

The urbanized site (site five) in Al Waab represents a dense existing urban development in Doha, with very little space for additional buildings, open space, trees, or other vegetation (Figure 2). It consists of primarily two-story residential buildings that are approximately six to eight meters in height, which are all behind a walled perimeter with many areas containing dense mature trees that are between 10 m and 12 m in height. The southeastern edge of the area is bordered mainly by covered parking structures that provide additional parking for nearby businesses and residents. All of the streets are two-lane residential roads with additional parking lanes on both sides. This site represents the typical building pattern in residential areas over the past 20–30 years.

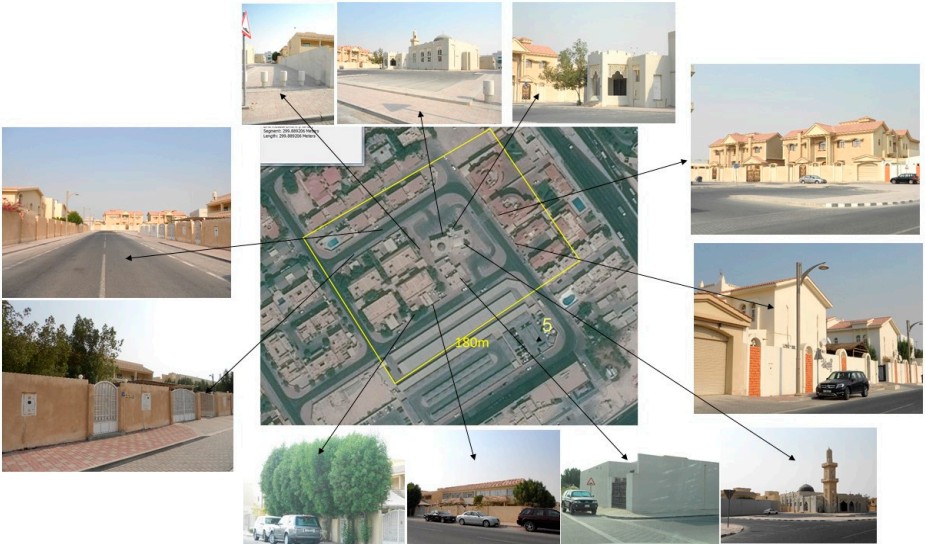

**Figure 2.** Photo survey of site five.

Finally, the coastal site (site nine) in Al Khulaifat represents a site with potential for redevelopment (Figure 3). The southern half of the site has a parking area, a mosque, and a small private group of low buildings (approximately six meters in height), and an area of very well-landscaped grounds in the southwest quadrant. The remainder of the site consists of a main road running east to west about halfway from north to south on the site, while the rest (northern half) is bare soil.

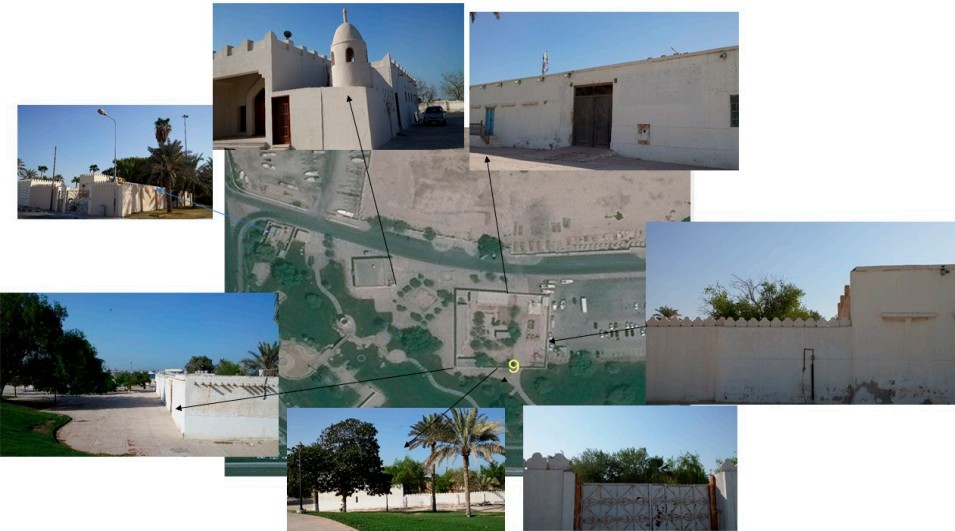

**Figure 3.** Photo survey of site nine.

## 2.2. Model Setup and Calibration

Each of the three selected sites was digitized in ENVI-met using a color-coded background image. The input configurations for the sites are shown in Figure 4.

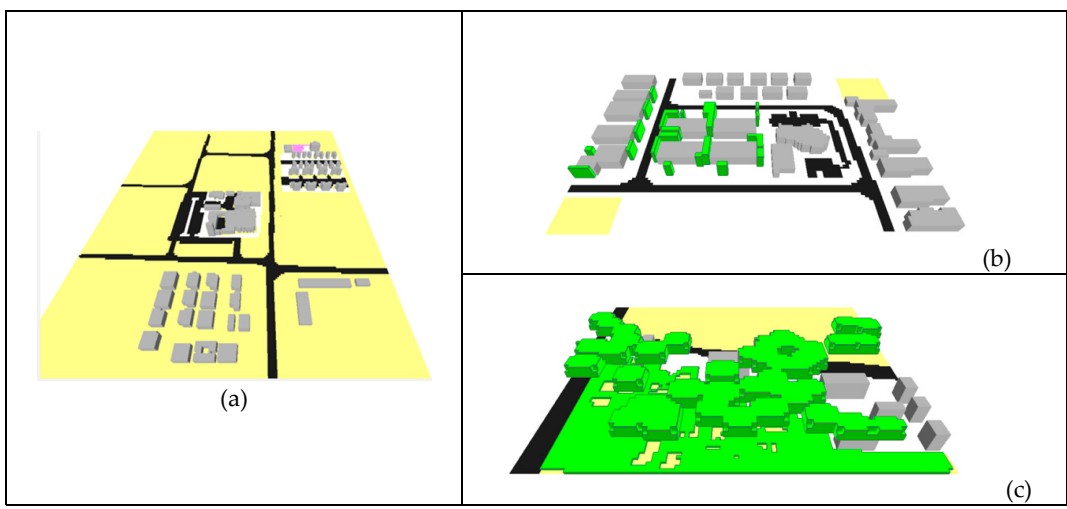

**Figure 4.** Color-coded representation of study areas in ENVI-met, (**a**) site one (Umm Salal Mohammad, developing), (**b**) site five (Al Waab, urban), (**c**) site nine (coastal) (gray = buildings, black = asphalt, yellow = sandy soil, green = vegetation).

The models were calibrated based on measured air temperature data from the weather station located on each site, and then analyzed for microclimate changes due to each change in material or vegetation. The measurement data served two purposes. First, the weather station provided empirical data for comparing air temperatures in different locations around Doha. Secondly, the data enabled the calibration of the ENVI-met model by setting a receptor in each model that corresponded to the weather station location. This approach aided in the validation of the modeling results. Model calibration is important when working with a model such as ENVI-met, as results can vary widely if the model is not calibrated to site-specific conditions (Figure 5). Appendix A shows the changes in initial parameters for each version of the base model calibration. Once a reasonable base model was obtained, one change was made in each scenario; then, results were compared to the base model. For this study, the hours of 06:00 to 18:00 were chosen, as this is the hottest part of the day, and the

timeframe for which the design improvements are likely to have the most impact on the comfort of residents.

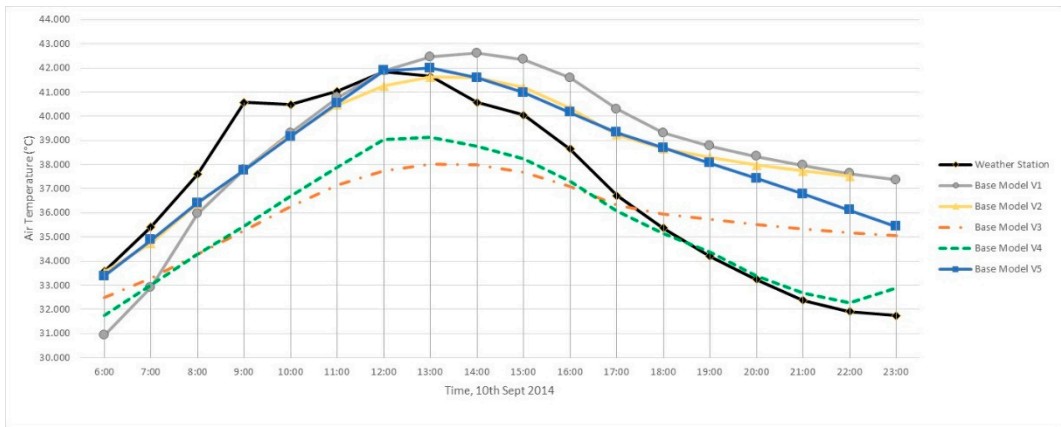

**Figure 5.** Calibration of site one base model, comparison of model results to weather station data for version one to version five (see Appendix A for model settings).

### 2.3. Modeling Scenarios

Following the setup of a base model for each site, which represents the current field conditions, a number of additional scenarios were tested in which one substantial change was made either to the materials, the vegetation, or the configuration of buildings. Various studies have advanced several key urban heat mitigation strategies, primarily: (1) urban forestry, (2) cool roofs, and (3) cool pavements [8,19,20,38,39]. Recently, blue space strategies (e.g., ponds, waterfronts, rain gardens, and water features) [12,40] and building arrangements [41–43] have also been investigated. In this study, we chose to assess realistic changes in building and landscape that could cool the urban environment in Qatar. Given that most buildings, especially residential, are already constructed of light-colored materials, including the roof, and that the arid environment is a somewhat limiting factor for water-based strategies, the remaining options to investigate include urban greening, changes to pavements, and changes to building arrangements. The model scenarios tested are described in Figure 6.

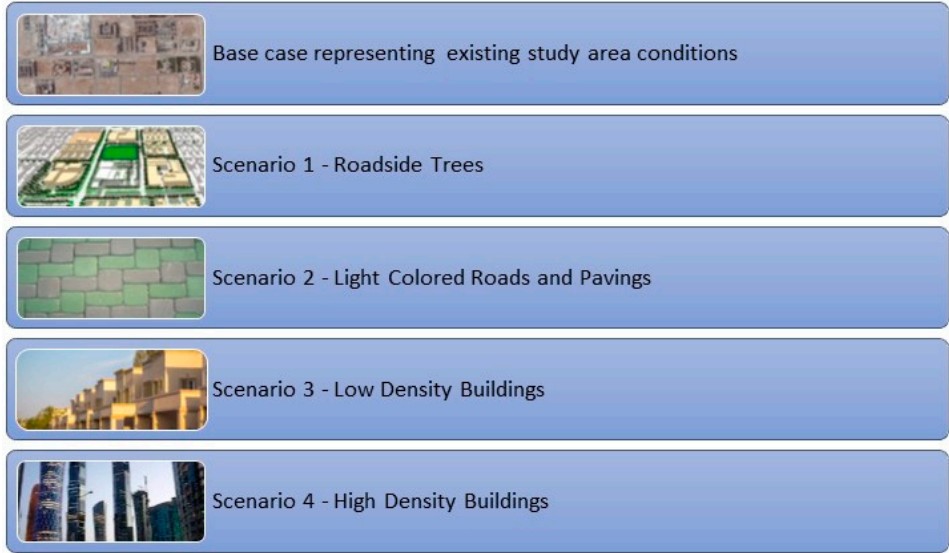

**Figure 6.** Scenarios tested in ENVI-met models (low-density photo credit: https://cdn.forbesmiddleeast.com/wp-content/uploads/sites/3/2016/10/ryad0002.jpg; high-density photo credit: http://whyqatar.me/images/tornado_tower__palm_towers.jpg; other images by authors).

While the range of output variables from ENVI-met is extensive, three key variables were selected to compare for this study. Studies of the UHI typically investigate changes in air temperature, surface temperature, or both, so these were initially selected. Mean radiant temperature ($T_{mrt}$) is typically considered as one of the most important variables related to thermal comfort under sunny conditions, and is defined as the uniform temperature of a hypothetical spherical surface surrounding the subject (emissivity $\varepsilon$ = one) that would result in the same net radiation energy exchange with the subject as the actual, complex radiative environment [44,45]. It is a required parameter in the calculation of several other thermal comfort indices, including PET (Physiologically Equivalent Temperature), the PMV (Predicted Mean Vote), UTCI (Universal Thermal Climate Index) and the PT (Perceived Temperature) [46]. In this study, we specifically wanted to compare both the effects of the selected changes on the UHI and on thermal comfort.

## 3. Results

### 3.1. Base Model Comparisons

The three base models were tested using a realistic set of input parameters until a base model that could reasonably approximate the measured air temperatures was developed. This section presents a comparison of the base model results to show overall thermal patterns on each site for the modeled day (10 September 2014).

### 3.1.1. Air Temperature

Table 1 presents the results of the daytime base model temperatures for the three sites at 06:00, 12:00, 14:00, and 18:00. In general, we find that the urbanized site (site five), has the lowest temperatures during the earlier parts of the day. However, this pattern changes after midday, and by 14:00, it is the warmest (with a maximum of 46 °C) and remains so until 18:00. The coastal site (site nine) shows similar minimum temperatures to the other two sites at 14:00, but the maximum temperature is slightly higher than that of the developing site (site one), and approximately 1.6 °C lower than that of the urbanized site. Overall, the daytime pattern of temperatures is similar for the developing (site one) and coastal (site nine) sites, but with the coastal site showing the highest maximum temperatures in the early morning, as expected for a site near a large water body.

**Table 1.** Base model daytime air temperatures (°C).

| Base Model Results Air Temperature—06:00 | Site 1—Umm Salal Mohammed (Developing) | Site 5—Al Waab (Urbanized) | Site 9—Al Khulaifat (Coastal) |
|---|---|---|---|
| Min | 32.44 | 29.81 | 31.89 |
| Max | 33.38 | 31.71 | 33.62 |
| Avg | 32.91 | 30.76 | 32.75 |
| **Base Model Results Air Temperature—12:00** | | | |
| Min | 40.46 | 39.85 | 41.37 |
| Max | 42.61 | 42.93 | 43.75 |
| Avg | 41.53 | 41.39 | 42.56 |
| **Base Model Results Air Temperature—14:00** | | | |
| Min | 41.92 | 42.21 | 42.32 |
| Max | 44.03 | 46.26 | 44.59 |
| Avg | 42.98 | 44.23 | 43.45 |
| **Base Model Results Air Temperature—18:00** | | | |
| Min | 39.29 | 40.23 | 39.13 |
| Max | 39.40 | 41.19 | 39.46 |
| Avg | 39.51 | 40.71 | 39.30 |

### 3.1.2. Surface Temperature

Focusing on the period of the day with most intense heat, the surface temperatures at 14:00 range from 33 °C to 63 °C, with site five (Urbanized) showing the highest maximum (Table 2). This is not much higher than site one (Developing), but is nearly 7 °C higher than site nine (Coastal).

At that time of day, site five is almost uniformly hot, at approximately 54 °C, with some areas that are approximately 47 °C between buildings, and small pockets of distinctly cooler surfaces at the western edges of buildings in the middle of the study area. Site nine (Coastal) has very slightly higher temperatures in the early morning, but remains distinctly cooler than the other two sites throughout midday. The coastal site (nine), which has mature trees on the southern half of the site, registers lower surface temperatures by approximately 2–3.5 °C on average during midday.

**Table 2.** Base model daytime surface temperatures (°C).

| Base Model Results Surface Temperature—06:00 | Site 1—Umm Salal Mohammed (Developing) | Site 5—Al Waab (Urbanized) | Site 9—Al Khulaifat (Coastal) |
|---|---|---|---|
| Min | 27.62 | 28.41 | 28.46 |
| Max | 32.85 | 32.85 | 33.19 |
| Avg | 30.24 | 30.63 | 30.83 |
| **Base Model Results Surface Temperature—12:00** | | | |
| Min | 32.85 | 32.85 | 32.85 |
| Max | 60.39 | 60.00 | 55.88 |
| Avg | 46.62 | 46.43 | 44.37 |
| **Base Model Results Surface Temperature—14:00** | | | |
| Min | 32.85 | 32.85 | 32.85 |
| Max | 62.01 | 63.13 | 56.27 |
| Avg | 47.43 | 47.99 | 44.56 |
| **Base Model Results Surface Temperature—18:00** | | | |
| Min | 32.85 | 32.85 | 32.85 |
| Max | 44.62 | 47.35 | 44.92 |
| Avg | 38.73 | 40.10 | 39.89 |

### 3.1.3. Mean Radiant Temperature

While mean radiant temperatures (MRT) do not show much difference within each site, they do show interesting differences between the sites (Table 3). The coastal site has a somewhat higher MRT in the early morning, especially compared to the developing site, but has the lowest MRT at midday: 7.7 °C cooler than the highly urbanized site, and over 4 °C cooler than the developing site at 14:00. Small areas, which have an MRT up to 20 °C lower than the rest of the site, are found on the east and northeast sides of vegetation and buildings at 14:00.

**Table 3.** Base model daytime mean radiant temperatures (°C).

| Base Model Results $T_{mrt}$—06:00 | Site 1—Umm Salal Mohammed (Developing) | Site 5—Al Waab (Urbanized) | Site 9—Al Khulaifat (Coastal) |
|---|---|---|---|
| Min | 20.30 | 21.53 | 23.38 |
| Max | 25.16 | 27.45 | 27.53 |
| Avg | 22.73 | 24.49 | 25.46 |
| **Base Model Results $T_{mrt}$—12:00** | | | |
| Min | 56.56 | 59.98 | 56.62 |
| Max | 77.67 | 81.11 | 73.54 |
| Avg | 67.11 | 70.54 | 65.08 |
| **Base Model Results $T_{mrt}$—14:00** | | | |
| Min | 58.18 | 61.21 | 55.73 |
| Max | 82.77 | 86.72 | 76.90 |
| Avg | 70.48 | 73.96 | 66.31 |
| **Base Model Results $T_{mrt}$—18:00** | | | |
| Min | 33.42 | 35.68 | 33.55 |
| Max | 34.94 c | 38.35 c | 35.44 |
| Avg | 34.18 c | 37.02 c | 34.49 |

### 3.2. Scenario 1: Roadside Trees

After calibrating and analyzing the base model for each site, the first scenario tested was the addition of trees, where trees were added at approximately 10-m intervals along the existing roads. The maximum air temperature of 1.2–1.3 °C was found for the developing and coastal sites (one and

nine), while the highly urbanized site (five), which has little available space for additional tree planting, showed almost no change (Figure 7). However, the highly urbanized site showed large reductions in surface temperature at 12:00 and 14:00 (up to 15°C), while the other two sites had 4–8 °C reductions during these times. Some small areas were found to have elevated surface temperatures, typically on the eastern side of trees in the late afternoon. This could be caused by the trapping of outgoing longwave radiation under tree canopies later in the afternoon and evenings. Interestingly, MRT was reduced by up to 19.52 °C at 14:00 for the developing site (one) and by similar amounts for the other two sites. Although MRT showed a slight increase across sites in the early morning, these results overall highlight the role of trees as an effective strategy for improving thermal comfort during the day with little or no drawback for evening or early morning hours.

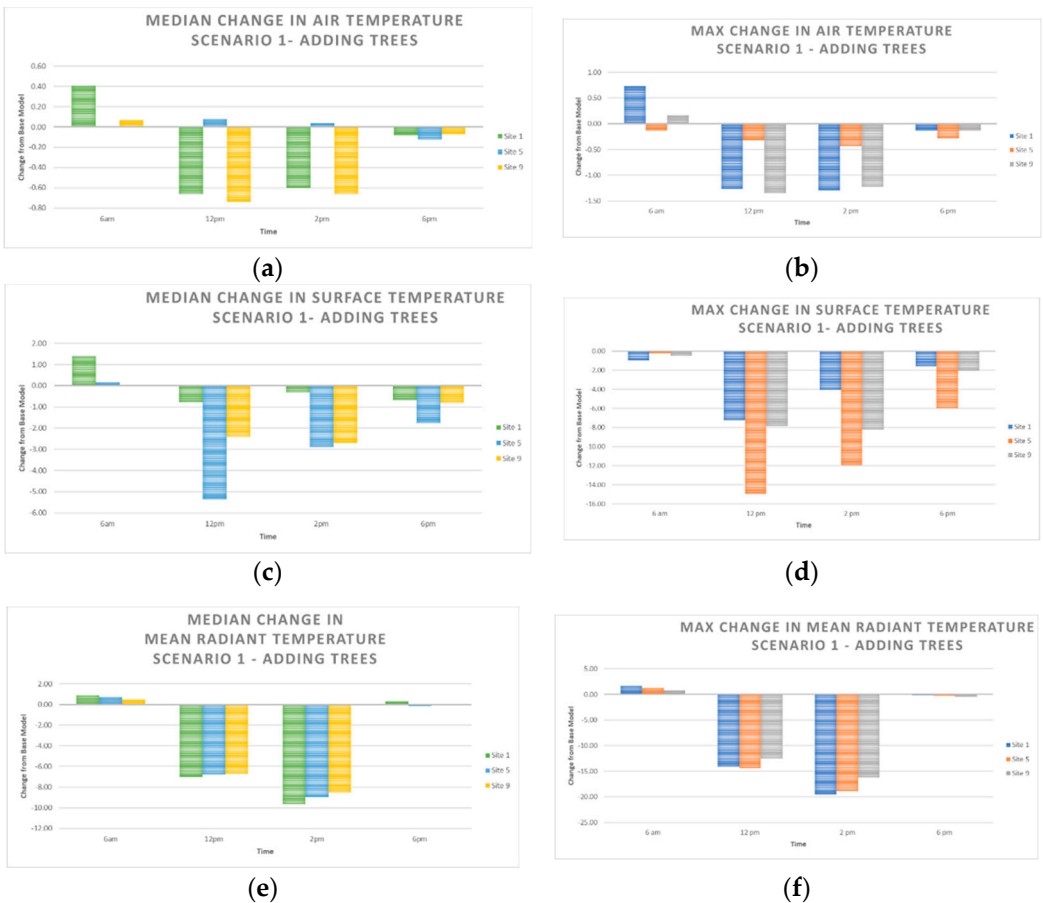

**Figure 7.** Median and maximum changes in air temperature (**a**,**b**), surface temperature (**c**,**d**), and mean radiant temperature (**e**,**f**) for Scenario 1: Roadside Trees.

### 3.3. Scenario 2: Lighter Roads and Pavers

For the scenario investigating roads and paving materials (changing to light gray pavers and a light-red asphalt), several observations can be made based on the results in Figure 8. It is primarily the surface temperatures that are improved (approximately 11 °C lower at midday for the coastal site 9 and 7.2 °C lower for the developing site one), while only small changes in air temperature (also 0.6–0.7 °C for the coastal and developing sites, and up to 2.9 °C for the urbanized site five, as in the scenario with added trees) are found. Surprisingly, the MRT increased on average for the urbanized and coastal sites (sites five and nine) during the daytime, while only very small reductions were found for the developing site. While the urbanized site (five) showed the largest average and maximum reductions in air and surface temperatures, it also showed an average increase in $T_{mrt}$ through a change in the roads and paving materials. As it is the site with the largest percentage of paved and asphalt

surfaces, the finding is significant, because it demonstrates that while near-surface air temperatures and surface temperatures may be reduced through a change in materials, the thermal comfort may be slightly worsened.

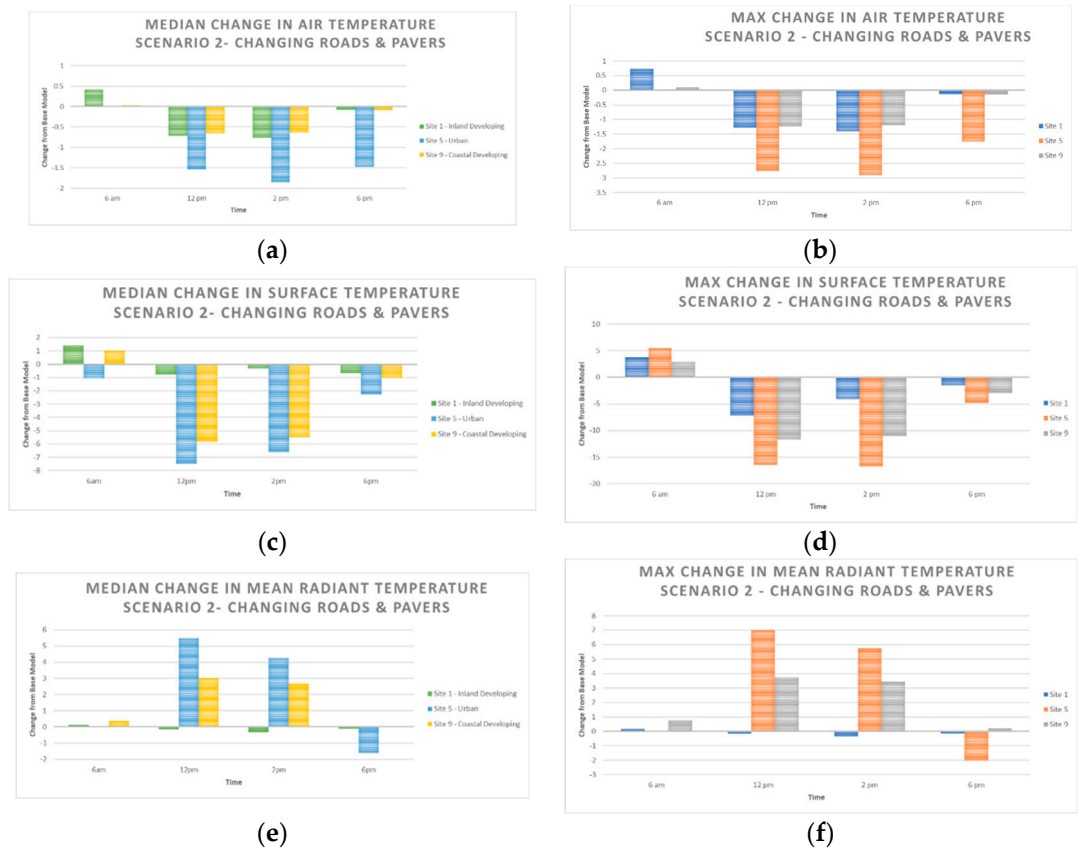

**Figure 8.** Median and maximum changes in air temperature (**a**,**b**), surface temperature (**c**,**d**), and mean radiant temperature (**e**,**f**) for Scenario 2: Changing Roads and Pavers.

### 3.4. Scenario 3: Low-Density Buildings

To test the effects of differing building arrangements, we first tested a scenario of a typical low-density building arrangement that is common in Qatar and many other countries in the region. Buildings in this arrangement are usually two to three-story detached or semi-detached residences arranged in a rectangular layout inside a walled plot. One compound can contain a small number of villas (20–30), or up to several hundred. As the developing and coastal sites (sites one and nine, respectively) are largely undeveloped, the building scenarios were tested on these two sites. The highly urbanized site (five), due its currently dense building arrangement, was not modeled for any alternative building arrangements.

While the changes are quite similar between the two sites across all times of day, some notable differences can be found at midday (Figure 9). Surface temperatures in early morning show higher maximum reductions for the coastal site (site nine), but this is mainly due to differences in small isolated patches at the north side of each building. At midday, reductions in both the $T_{mrt}$ and surface temperature are found on both sites, but again, these are in small strips around the northern and eastern edges of the buildings, due to slight building overshadowing at that time of day. Other areas between the buildings and on the southern side show increases of approximately 2.6 °C for surface temperature and almost no change for $T_{mrt}$. While the patterns are similar in terms of locations of increases and reductions for both sites, the absolute change appears to be greater for the developing site (site one) than for the coastal site (site nine). Interestingly, in the developing site, in which we tested several areas of courtyard-style building arrangements, compared to semi-detached rows of

buildings, the interiors of courtyards show reductions of approximately 6 °C for surface temperature, while the semi-detached rows showed increases of approximately 9 °C on roads. Air temperature reductions, which were typically between 1 °C and 1.5 °C, were more uniform across the site, and depended on the localized air temperature.

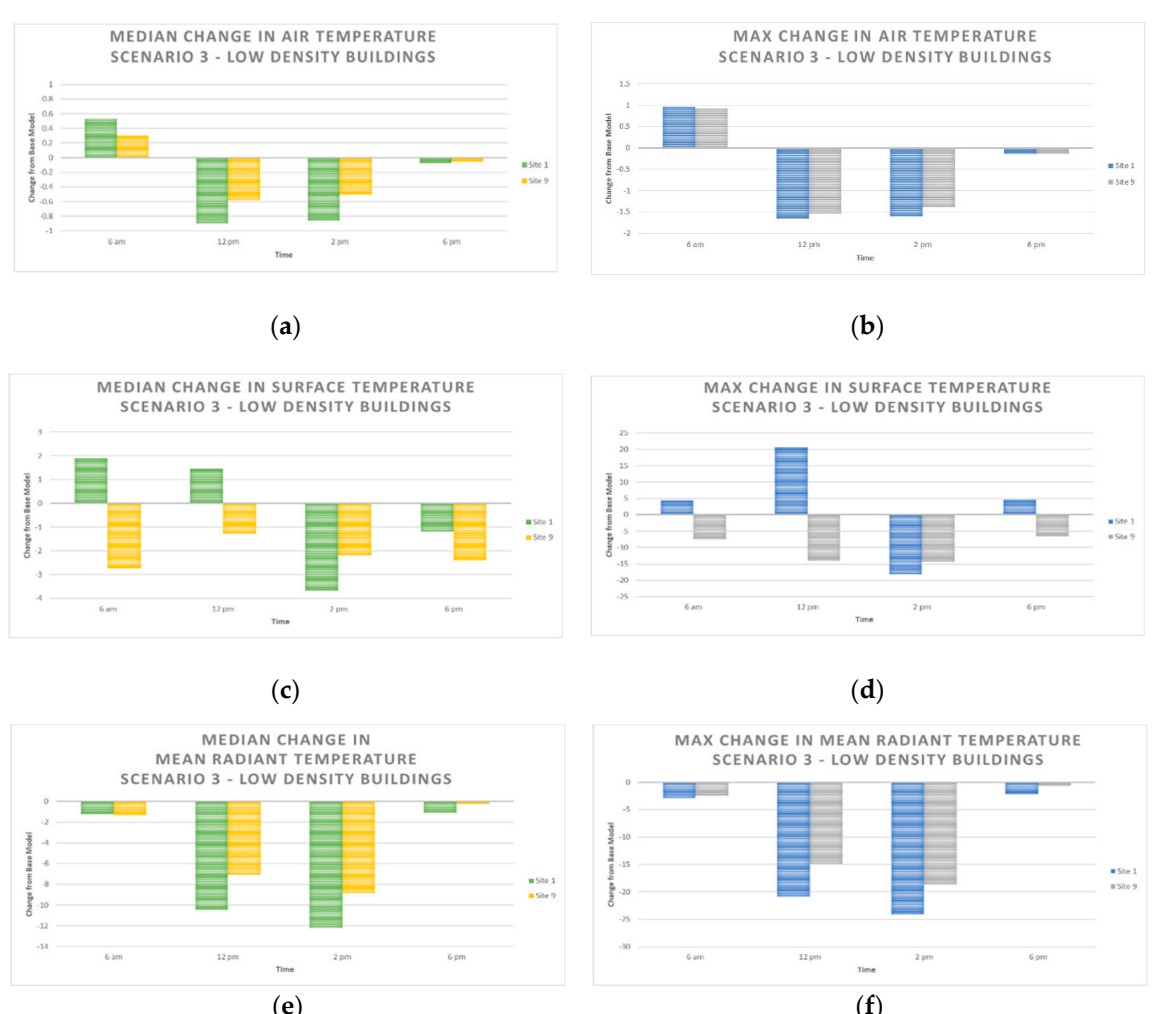

(**a**)          (**b**)

(**c**)          (**d**)

(**e**)          (**f**)

**Figure 9.** Median and maximum changes in air temperature (**a**,**b**), surface temperature (**c**,**d**), and mean radiant temperature (**e**,**f**) for Scenario 3: Low-Density Buildings.

*3.5. Scenario 4: High-Density Buildings*

For the high-density building scenario, we tested the sites with tower-block style arrangements of square buildings of 30-m height. Both sites, developing and coastal, showed large reductions in all parameters, especially surface temperatures and $T_{mrt}$ during midday, with site one showing the largest reductions at 12:00 and 14:00 (Figure 10). However, it should be noted that these areas of large reductions are limited to small strips in the shadows of buildings, as in the scenario for the low-density building arrangements. All of the areas of paved surfaces (either asphalt or paving blocks) showed increases of surface temperature and $T_{mrt}$ of 4–8 °C at midday. The temporal patterns of temperature change are quite similar between low density and high density, with slightly greater reductions in air temperature and $T_{mrt}$ in the high-rise scenario at midday in the developing site (site one), and for all three parameters in the coastal site (site nine).

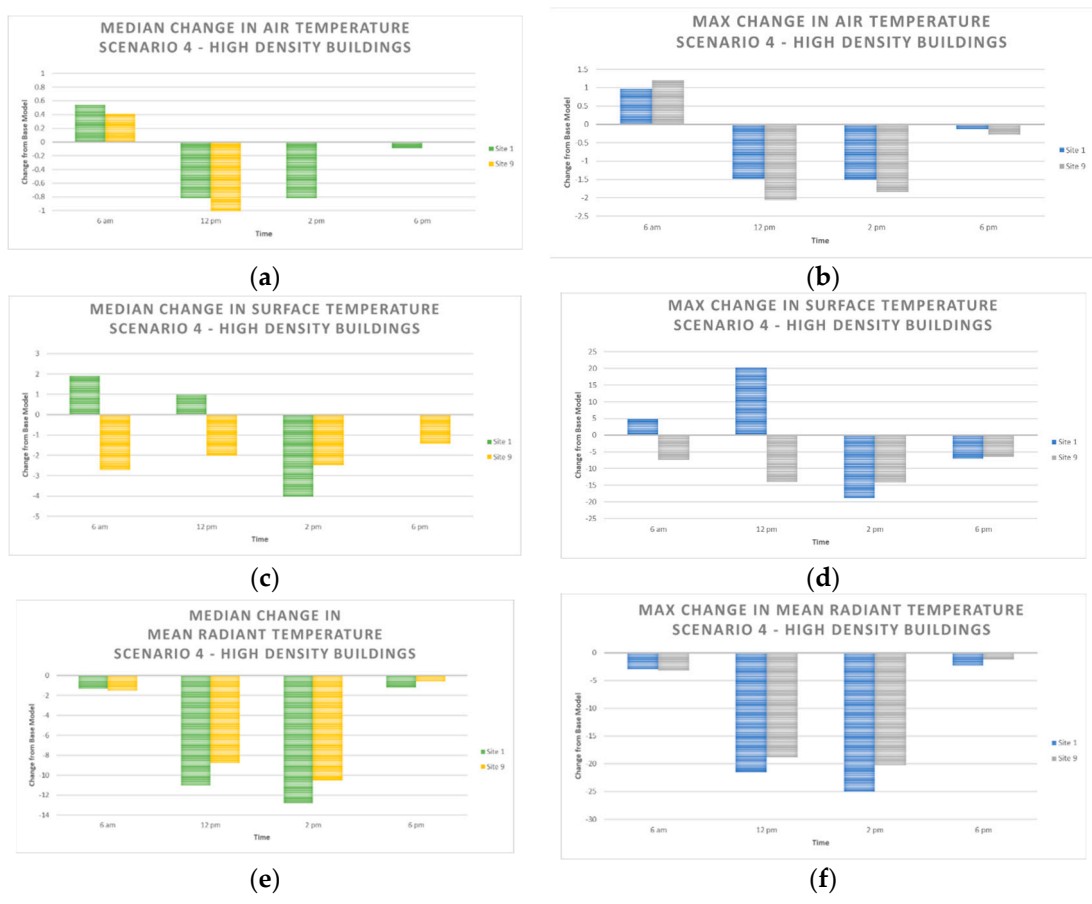

**Figure 10.** Median and maximum changes in air temperature (**a**,**b**), surface temperature (**c**,**d**), and mean radiant temperature (**e**,**f**) for Scenario 4: High-Density Buildings.

## 4. Discussion

The four scenarios presented here provided comparisons of air temperature, surface temperature, and mean radiant temperature (an indicator of thermal comfort) between the base model and the models that have been changed with: (1) additional tree cover along roadways; (2) higher albedo roads and pavements; (3) low-density building arrangements, and (4) high-density building arrangements. This section explores the most effective strategies related to the parameters investigated.

### 4.1. Air Temperature Reduction Strategies

For air temperature near the ground (one-meter height), the model scenario with added trees was lower by approximately 0.6–0.7 °C, depending on the existing temperature regime in that area, for the developing and coastal sites (sites one and nine, respectively). For the urbanized site, the air temperature reductions were very minimal, but this is reasonable, considering that the site had fewer available areas for planting new trees, and also that the change was from a paved/asphalt area to a shaded paved/asphalt area. For the developing and coastal sites, the change was from bare soil to shaded bare soil. In the highly urbanized site five, changing to lower-albedo pavers and roads also demonstrated a good reduction in air temperature (up to 2.9 °C) at 14:00. The developing and coastal sites showed maximum reductions of 1.2 °C to 1.4 °C. For building changes, the high-density building scenario showed the strongest improvement for air temperature, but only in small areas in the afternoon (12:00 and 14:00) shadows of buildings. For the developing site, slightly greater reductions in air temperature were found in the low-density building scenario. Other research in the temperate climate of the United Kingdom (UK) has found similarly modest reductions in air temperature due to shading, 0.1–0.2 °C for a 5% increase in the tree cover across a suburban area [37], but up to 0.9 °C

for shaded areas compared to open areas of a park [47]. A meta-analysis by Bowler et al. [48] that investigated the temperature differences between parks and their urban surroundings estimated an average air temperature reduction of 0.94 °C by synthesizing results from a large number of studies from many different geographic locations and climatic regimes.

*4.2. Surface Temperature Reduction Strategies*

In examining the surface temperature differences, we find that the areas immediately under trees have surface temperatures that are reduced by up to 15 °C in urbanized site five, while the other two sites showed average reductions of 0.7 °C to 2.7 °C at midday. Urbanized site five represents a change from pavers or asphalt to a shaded paved surface, whereas the developing and coastal sites represent a change from bare soil to shaded bare soil. Although the low-density and high-density buildings show large maximum reductions in surface temperatures, these are limited to small areas around the edges of buildings. Therefore, the average surface temperature reductions across the whole site are actually greatest in the urbanized site, which has the largest amount of urbanized surface that is being changed to higher albedo materials.

While the scenario of changing the type of paver alone resulted in approximately 2 °C lower surface temperatures compared to the base model, we also found in comparing different areas of the base model that a standard light red paver will be lower by up to 12 °C than areas covered with asphalt and 3 °C to 4 °C cooler than areas of bare soil.

Only a few studies have attempted to analyze changes in surface temperature over time for different surface types. A recent study found that changing surfaces from concrete to grass reduced the surface temperature by up to 24 °C [47] in a UK site, whereas another study in Basel, Switzerland [49] that used a high-resolution thermal camera to take an aerial image estimated that streets were approximately 11 °C warmer than vegetated areas. Using modeling studies, Kjelgren and Montague [50] found that afternoon asphalt surface temperatures ($T_s$) were between 20 °C and 25 °C higher than turf $T_s$, whereas Gill [51] found a maximum difference of 25°C between non-transpiring surfaces such as concrete and wholly transpiring surfaces such as woodland across Greater Manchester, UK.

*4.3. Mean Radiant Temperature Reduction Strategies*

Finally, in examining the results for mean radiant temperature, all of the sites showed large reductions for $T_{mrt}$ under the scenario of additional trees. Mean radiant temperature is a useful proxy for thermal comfort, as it essentially measures the mean of all of the surface temperatures in a given space, and the thermal exchange with a particular point (e.g., a human body). The areas underneath trees are cooler by up to 19.5 °C at midday. While this is quite a significant result, it should also be noted that trees do have the potential for maintaining slightly higher $T_{mrt}$ than the surrounding areas during the evening and nighttime due to the trapping of outgoing longwave radiation under the tree canopy. The models for this area show slightly elevated nighttime temperatures of 0.4–0.7 °C at 22:00. This is a seemingly small trade-off when compared to the large daytime improvements in thermal comfort achieved with trees. During the day (at noon), $T_{mrt}$ can reach up to 72 °C, while at 22:00, it is around 30 °C across the study area.

It is also worth noting that $T_{mrt}$ is increased or shows almost no improvement across all of the sites for the scenario of changing pavers and roads. This is interesting in that increasing the albedo (using light-colored surfaces) is often cited as a strategy for alleviating urban heat. However, while this certainly leads to an improvement in surface temperatures, it may also reduce the near-surface thermal comfort by increasing the $T_{mrt}$ near the ground surface [52]. This result is supported by previous research showing that sheltered locations (either sheltered by vegetation or by surrounding buildings) show greatly reduced $T_{mrt}$ as compared to exposed locations [47,53].

*4.4. Limitations*

A number of limitations to this study should be noted. These include: (i) as stated in Section 2.3, not all of the relevant interventions were tested here, because light-colored buildings and roofs are already standard building practice, and the arid climate places some limitations on water-based strategies; (ii) we selected only three neighborhoods in Doha for modeling due to the extensive time requirements for running models; therefore, some neighborhood configurations of Doha may not be represented here; and (iii) while human thermal comfort depends on both indoor and outdoor temperatures, the UHI may also lead to the increased use of air conditioning, which leads to greater carbon emissions unless non-fossil fuel renewable energy sources are used; while examining this is beyond the scope of this study, it should be noted that attempts to reduce the UHI and improve thermal comfort are closely tied to climate change mitigation.

## 5. Conclusions

Considered together, the four scenarios presented in this research point toward a number of planning guidelines that can be adopted at the neighborhood level for hot arid regions. Although the research may not be able to specifically determine how the physical changes are affecting Doha residents in terms of health and comfort, it describes the temperature patterns in three case study areas, and provides guidance on alleviating urban heat, depending on the thermal regimes in the different areas of Doha.

One consideration is the temporal variation in temperatures associated with differing types of land cover. This may lead to opportunities for reducing temperatures during "shoulder periods", or transition times, during late morning and early evenings, which may offer chances for changing the urban design such that residents are able to spend more time outside. If planning agencies are considering options for mediating temperatures to provide pedestrians greater access to outdoor spaces, then reducing the amount of urban land cover (or impervious surfaces) may be a first step. While changing land cover may not be cost-effective or a feasible option in places containing a large amount of impervious surface, making space for trees within highly urbanized areas may be a reasonable alternative. Despite the arid climate, given the abundant amount of water from reusable sources such as air-conditioning condensate [54] and treated sewage effluent (TSE) in Doha [55], water resources for expanding the urban canopy may be readily available.

Although increasing the albedo of surfaces is a common practice that could reduce the absorption of solar radiation, it might not work for Doha and places with similar climatic conditions. Higher albedo could help the temperature in the morning and evening, but it could also have the adverse effect of increasing both the near-surface air temperature and mean radiant temperature during the daytime.

Finally, while the high-density, high-rise development did estimate large surface temperature and $T_{mrt}$ reductions at midday, this only occurred in small areas directly in the shadow of buildings. It is important to consider that if coastal winds are blocked by high-rise buildings along the coastline, inland areas may not benefit from this mediating influence. Restricting development along the coast, especially those buildings that prevent these coastal processes from mediating inland temperatures, is a policy that has traction in scholarly research [56], and it may be a policy option that can improve the short-term and long-term quality of life of Doha's residents. Also, the research indicates that lower density development may be the best solution for this climate, in that it that allows for air circulation and shading between spaces through the use of natural, soft-scaped, communal spaces between buildings and paved areas.

Again, at the neighborhood scale, the most effective scenario was that of adding mature trees (approximately 10 m in height, in 10-m intervals along the sides of roads). In the coastal site, the microclimate modeling estimated a maximum hourly air temperature reduction of 1.35 °C and surface temperature reductions in site five of up to 15 °C at 12:00. While the scenario of changing the type of paver resulted in approximately 3 °C lower air temperatures compared to the base model for the urbanized site, we also found that comparing different areas of the base model showed that a standard

light red paver will be lower by up to 12 °C than areas covered with asphalt, and 3 °C to 4 °C cooler than areas of bare soil. However, $T_{mrt}$ was also increased or nearly neutral across all of the sites for the pavers and roads scenario. This result points to the need to develop improved shading measures for pedestrian pathways and outdoor recreational areas, especially for inland areas, as these experience the strongest UHI at midday when both air temperatures and $T_{mrt}$ are highest.

This study suggests a need for further and ongoing examination of Doha's urban heat and development patterns, particularly to gain additional insight on the specific planning measures that are most likely to lead to improvements in the thermal comfort of residents. Such future research will continue to inform the planning choices, not only for Doha, but for the wider Middle East and North Africa region, and help reduce the possible negative implications of development for humans and the environment upon which they depend.

**Author Contributions:** Conceptualization, V.S. and S.F.; methodology, V.S., S.F. and C.S.; software, C.S. and Y.M.; validation, C.S., V.S., and Y.M.; formal analysis, C.S. and V.S.; investigation, V.S. and S.F.; resources, S.F.; data curation, C.S.; writing—original draft preparation, S.F., C.S. and V.S.; writing—review and editing, V.S. and Y.M.; visualization, C.S. and Y.M.; supervision, S.F. and V.S.; project administration, S.F.; funding acquisition, S.F. and V.S.

**Funding:** This research was conducted under the NPRP grant # NPRP 5-074-5-5015 from the Qatar National Research Fund (a member of Qatar Foundation).

**Acknowledgments:** Thanks to Aya Nafi, research assistant at Qatar University, who produced Figures 4–6 and extracted much of the ENVI-met results for analysis by the research team. Also, the Portland State University team was supported by the Institute for Sustainable Solutions.

**Conflicts of Interest:** The authors declare no conflict of interest. The funders had no role in the design of the study; in the collection, analyses, or interpretation of data; in the writing of the manuscript, or in the decision to publish the results.

# Appendix A

**Table A1.** Model Settings for Base Model Calibration.

| Model Version | Wind Speed (10 M) | Initial Temp | Spec Hum | Rel Hum | LBC (Lateral Boundary Condition) | Soil Wetness | Soil Initial Temp | Solar Adjustment | Time Steps | Switching Angles | Timing (Plant, Surface Data, Radiation) | Flow Fields |
|---|---|---|---|---|---|---|---|---|---|---|---|---|
| **Base V1** | 2 | 310 | 4 | 40 | Open | 30, 40, 40, 60 | 306, 304, 300, 293 | none | 20,10,5 | 30, 50 | 900,60, 900 | 1800 |
| **Base V2** | 4 | 310 | 4 | 40 | Open | 30, 40, 40, 60 | 306, 304, 300, 293 | none | 2,2,1 | 40,50 | 600,30,600 | 900 |
| **Base V3** | 5 | 308 | 6 | 45 | Open | 30, 40, 40, 60 | 306, 304, 300, 293 | 0.7 | 2,2,1 | 40,50 | 600,30,600 | 900 |
| **Base V4** | 4 | 307.39 | 4 | 45 | Simple Forcing | 30, 40, 40, 60 | 306, 304, 300, 293 | 0.9 | 2,2,1 | 40,50 | 600,30,600 | 900 |
| **Base V5** | 4 | 310 | 4 | 45 | Simple Forcing | 30, 40, 40, 61 | 306, 304, 300, 293 | 0.9 | 2,2,1 | 40,50 | 600,30,600 | 900 |

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
