# Peer review of "A Comparison of Neighborhood-Scale Interventions to Alleviate Urban Heat in Doha, Qatar"

_sustainability, doi:10.3390/su11030730_

Round 1
Reviewer 1 Report
Comments
1. This article deals with a very important theme on the urban thermal environment, and it is very interesting content. However, descriptions on research methods are inadequate, in particular, there are few descriptions such as three dimensional calculation domain of CFD model as the base model, boundary conditions, tree modeling, turbulence model etc. Therefore, it is difficult to evaluate the validity of the calculation results.
2. In general, a comprehensive thermal environment indicator such as SET * (Standard New Effective Temperature) is often used as an evaluation index of thermal comfort, but in this paper, the evaluation is done using air temperature, surface temperature, MRT. You should specify the reason why you used the indicators.
3. The legend of Figure 5 is meaningless. It should be noted what Base V 1 to V 5 represents. From this result it cannot be understood whether the validity of the base model can be verified.
4. The values, air temperature, surface temperature and MRT, in Tables 1-3 are unknown at which points in the base models.
Author Response
Please see attached Word document for responses.

Reviewer 2 Report
This manuscript presents the results of microclimate modeling of four types of urban heat-alleviation intervention in three neighborhoods of Doha, Qatar which differ in their urbanization and built materials. The authors state their aim as understanding “which physical features of urban design can improve thermal comfort in outdoor spaces during periods of intense heat” in the city of Doha. The study tracks three main outcome indicators related to thermal comfort: air temperature, surface temperature and mean radiant temperature.
The authors’ rationale for the study rests on the well-accepted view that the most appropriate heat island reduction actions are location-specific, and on the extreme heat emergency facing the cities of the Gulf region. Knowledge is urgently-needed regarding which of a number of well-known heat-alleviation interventions are most effective in reducing risk to populations in high-risk cities (or city neighborhoods) given vast differences in built environment and landscape characteristics and the nature of heat island challenges across (and within) world cities. The study contributes to the field by providing location-specific results for Doha that may also be relevant for other arid coastal cities, as well as demonstrating a replicable methodology. Congratulations to the authors on an interesting study, and best of luck with this research.
My comments are provided by MS section below. The three main comments are (i) the MS would benefit from greater clarity on the four sets of interventions, three study areas, and three sets of outcome indicators – particularly in the context of the stated study goal; (ii) it would be useful to further detail in the MS the human health impacts of extreme heat (once again, in light of the study goal of human thermal comfort); and (iii) a discussion of study limitations is needed, including why the particular interventions were chosen (and others were not).
Abstract
Line 24-25. Reference should be made in this last sentence to Doha. Currently it reads as a general recommendation for any city, and I think the point of the study is to conclude on specific recommendations for Doha, and perhaps other arid coastal cities with Doha’s built environment and climate challenges.
Introduction
Lines 41-44. It would be useful here to provide for readers the simple definition of “urban heat island effect”
Lines 51-53. What is missing here: “stave off massive human population health impacts.”
Line 57. “Increasingly…”
Line 86. “RCP8.5…”
Line 88. Heat (or sun) stroke is only one of many health impacts of extreme heat. Others include increased risk of cardiovascular disease, worsening of respiratory conditions, risks to kidney function, risk of adverse birth outcomes, etc. Also, not mentioned is reduced labor productivity which is also already being seen in Gulf countries and elsewhere and which has a direct economic as well as wellbeing impact. The context for the study – “thermal comfort and livability” – fundamentally drive human health and wellbeing. There is a large literature on heat-health outcomes including the 2018 Lancet Countdown study.
Starting line 90. To best contextualize study findings, in this para it would be useful to provide a bit more background on Doha’s built environment and landscape, e.g., seacoast, historic sections with traditional (light-colored, densely built) neighborhoods, recently-built high-rises. Some of this provided throughout several paras but would be useful to consolidate in one place.
Line 100. Something is amiss in the sentence.
Line 101-102. Is the reference to Doha’s design like northern hemisphere cities not more appropriate in previous para? To fully understand the last sentence of this para, the additional background on Doha’s built environment noted above would be helpful.
Methods
Starting line 133. The reference to the study sites by weather station number – 1, 5 and 9 – is confusing for those who are not familiar with the study. It would be clearer to use the terms “urban,” “coastal” and “urbanizing” (or some other appropriate words) throughout to label the three study sites – as in fact is done in the tables.
It would be important to add a sentence explaining why these three sites were chosen, in addition to the fact that they are diverse (e.g., I assume they are representative of some of the typical built environment/landscape contexts to be found in Doha?…)
Starting line 147. There is some repetition in this para when speaking of little space for additional buildings, trees, etc. Check other parts of the MS also for some repetition which could be reduced to enhance readability (and some typos).
After line 184. This section requires some sort of rationale for choosing these specific interventions. Why these, and not others? (e.g., why not increasing impervious surfaces, rain gardens, courtyard gardens, white walls, white roofs, etc.) I am sure that there are good reasons, related to appropriateness to the local arid context, experience in similar urban/climate circumstances etc. However, currently no rationale for these particular interventions/scenarios is provided, and should be. Ideally, this should be based at least in part on the literature.
Line 187. URL is missing.
Figure 6. This would be clearer if displayed here in the same way described in the paper, i.e., as “four scenarios.” Currently it looks like a base case plus five interventions. As with the comment above for the study sites, it would help to reduce confusion for readers if the four scenarios were labeled with short terms, and those terms consistently used throughout – e.g., “1. roadside trees,” “2. lighter surfaces,” “3. low-density buildings,” “4. high-density buildings” or some other such terms.
Generally, the photos and images provide a helpful and effective aid to comprehension of the study.
Missing from this Methods section is a brief explanation of the three outcome measures used in the study – air temperature, surface temperature and mean radiant temperature. The latter, in particular, requires a definition in the Methods section because it is not a commonly-used measure in other sustainability-related science fields. In particular a short explanation of why MRT was chosen and its link to the thermal comfort objective noted in the goal statement (line 116) is warranted. Some of this information is provided currently in lines 347-349 but it comes too late in the MS, it should be moved up to Methods.
Results
Some of the detail is probably not needed to fully convey the findings; this and some repetition contribute to making the findings somewhat confusing to sort through. The clear and consistent labeling of sites and scenarios noted above will help. Shortening a bit could help also.
Discussion
Line 319. Other research where? As noted in this MS, location is very important for relevance!
Line 322. Other studies where?
After line 337. As above, where?
This Discussion section needs a brief para on study limitations. Among points that could be addressed: (i) not all relevant interventions were tested here, and explain why; (ii) some neighborhoods of Doha may have different requirements and these were not able to be tested; (iii) human thermal comfort depends on both indoor and outdoor temperatures, and the urban heat island effect influences the indoor temperature as well (including use of air conditioning, which worsens global warming unless non-fossil fuel renewable energy sources are used); this was beyond the study scope, but it should be mentioned in some way; (iv) some of the points currently found in the Conclusion section are probably more appropriate to the Discussion section (see below).
Conclusions
After line 374. This para is more appropriate to the Discussion section.
After line 390. As above, this para may also be more appropriate to the Discussion section.
Author Response
Thank you for the very constructive comments. Please see the attached document for the responses to Reviewer 2's comments.
